# Effects of Muscle Strength, Agility, and Fear of Falling on Risk of Falling in Older Adults

**DOI:** 10.3390/ijerph20064945

**Published:** 2023-03-11

**Authors:** Filipe Rodrigues, António M. Monteiro, Pedro Forte, Pedro Morouço

**Affiliations:** 1ESECS—Polytechnic of Leiria, 2411-901 Leiria, Portugal; 2Life Quality Research Center, 2040-413 Leiria, Portugal; 3Department of Sport Sciences and Physical Education, Polytechnic of Bragança, 5300-253 Bragança, Portugal; 4Research Centre in Sports Sciences, Health, and Human Development, 6201-001 Covilhã, Portugal; 5ISCE Douro, 4560-708 Penafiel, Portugal; 6Center for Innovative Care and Health Technology (CiTechcare), 2410-541 Leiria, Portugal

**Keywords:** elderly, muscle strength, accidental fall, fear

## Abstract

Falls are a major public health problem among older adults because they lead to premature mortality, loss of autonomy, and increased dependence on others. However, these associations have not been explored using procedures that analyze the sequential effects between risk factors of falling. The present study aimed to examine the effects of muscle strength, agility, and fear of falling on risk of falling using path analysis in community-dwelling older adults. In total, 49 elderly (female = 33, male = 16) participants aged between 65 and 76 years (M = 68.38 years; SD = 6.22) were included for analysis. Muscle strength, agility, fear of falling, and risk of falling were assessed using validated instruments for the older adult population. The proposed model shows that muscle strength was negatively associated with agility. Consequently, agility was negatively associated with fear of falling. The same trend appeared between fear of falling and risk of falling. The effect sizes were between small and medium for agility (*R*^2^ = 0.16), fear of falling, (*R*^2^ = 0.29), and risk of falling (*R*^2^ = 0.03). The main finding of the present study was that muscle strength was significantly correlated with agility, which, in turn, predicted fear of falling. Consequently, lower scores for fear of falling explained lower risk of falling in community-dwelling older adults. While muscle strength is a crucial component of physical fitness, only with adequate levels of agility can older adults possess the efficacy and ability to perform daily tasks.

## 1. Introduction

Falls are a global public health problem. Research [1,2] has reported that one third of older adults aged over 65 years fall each year. The ratio of injuries caused by falls increases with age, affecting up to 40% of elderly people aged over 75 years, and 50% of elderly people aged over 80 years [3]. The consequences of falls in the elderly are devastating. About 15% of falls result in dislocations, contusions, and muscle trauma [4], and about 10% of falls result in bone fractures [5]. As described in the literature, fractures associated with falls in the elderly are a significant source of morbidity and mortality [5,6]. In addition, falls have psychological consequences, such as loss of confidence and increased fear of falling, which can result in restricted activities, leading to reduced physical functions and social interactions [7]. Studies conducted by several authors [1,8] show that falls in the elderly have been increasing in the last 10 years, and the implementation of efficient strategies to reduce both the risk of falls and the incidence of falls could reduce the costs associated with healthcare [4,9]. In addition, the identification of correlates for risk of falling is of utmost importance as a means to provide quality of life in older adults.

Muscle strength is the main component of physical fitness, since only with acceptable levels of strength can the elderly perform different daily life activities, such as climbing stairs, shopping autonomously and safely [10,11], and leisure-time physical activity [12]. With aging, muscle loss and muscle strength are responsible for major functional limitations and their associated morbidities [13]. One consequence of sarcopenia is a decrease in agility and mobility, consequently increasing the risk of falling [4]. Hence, elderly people who show greater scores for muscle strength have been shown to take less time to perform agility tests [14], suggesting greater levels of performance in daily tasks with adequate autonomy and functional capacity. In this regard, since agility seems to be related to balance, its association with fear and risk of falling seems to be a research topic that should be explored [15].

Agility and balance are defined as the speed and positioning with which a person performs a given movement. Rodrigues et al. [4] state that the ability of the body to maintain balance depends on several factors, including the central nervous system and muscle strength. Thus, greater muscle mass should be associated with greater ability to move, since for appropriate balance, the elderly must have the strength to perform a given movement and the coordination to remain maintain it for any length of time [16]. Thus, elderly people with greater agility would have a lower risk of falling, considering the muscle strength associated with agility and balance [4].

### Current Study

According to several authors [5,17], intrinsic factors of falls, such as impaired vision, reduced lower extremity strength, and reduced grip strength, have been associated with risk of falling. Additionally, performance-based assessments of gait and balance have found significant associations between dynamic balance and falls [2,18]. Considering these intrinsic factors, there is evidence that determinants of falls are mostly related to internal variables related to the elderly. These factors increase the likelihood of producing fear of potential falls in the near future. However, it is of note that fear and risk of falling are two distinct factors. Fear of falling (or a lack thereof) is related to the psychological level of individuals with regard to performing daily activities. On the other hand, risk of falling is related to the ability or inability to safely perform tasks with adequate levels of balance. Greater levels indicate balance capacity, and thus, lower risk of falling. Fear of falling increases the risk of falling [19], and thus, it is also expected that lower fear of falling indicates greater balance abilities in older adults.

Regarding falls in older adults, fear of falling should not be underestimated. It assumes a concern about falling as a consequence of loss of agility and balance, loss of confidence, and avoidance of daily activities [20]. Moreover, several authors [17,21] have indicated that fear of falling could predict risk of falling among the elderly. Numerous studies have found a link between fear of falling and risk of falling in older persons [17,19,22]. The strength of the link, however, varies between groups based on the sample characteristics, measurement instruments, and statistical methodologies used. Furthermore, the determinants that predict the association between fear of falling and risk of falling are still a topic of research interest [1,4]. Research is needed to understand the mechanisms underlying the link between fear of falling and risk of falling, as well as the specific components that contribute to this link [23]. Sequential modeling can be useful for developing a deeper insight into the relationships between the determinant factors and the role they play in the risk of falling. Despite the extensive epidemiological research on risk factors associated with falling, there is a lack of detailed information on how functional fitness can predict fear and risk of falling. That is, identifying the sequential path of risk factors for falls may be crucial in planning fall prevention programs. The present study aimed to examine the effects of muscle strength, agility, and fear of falling on risk of falling in community-dwelling older adults. Our initial hypothesis was that muscle strength would indicate greater scores for agility, and consequently, decreased levels of fear of falling. Ultimately, we expected that decreased fear of falling would indicate lower risk of falling in older adults.

## 2. Materials and Methods

### 2.1. Participants

Prior to conducting the study, a sample size calculation for path analysis was performed to determine the necessary sample size to achieve adequate statistical power. The minimum acceptable anticipated effect size was 0.3, desired statistical power was 0.95, number of observed variables was 4, and probability level was 0.05, which indicated that the calculated minimum sample size was 44.

This cross-sectional study was conducted in a higher education institution. Data collection procedures during a 2-week period were performed. In total, 49 older adults (female = 33, male = 16) aged between 65 and 76 years (M = 68.38 years; SD = 6.22) were included for analysis. The potential participants were carefully informed about the design of the study, and information was provided regarding the possible risks and discomfort related to the undertaking of physical fitness tests. Informed consent forms were read and signed by each individual prior to study participation. For inclusion in the study, we considered those who met the following inclusion criteria: (i) those aged 65 years or older; (ii) those who had provided informed consent to participate; and (iii) those who had undertaken more than six months of regular exercise.

### 2.2. Procedures

This study was approved by the Scientific Board of the Higher Institute of Educational Sciences of the Douro (n° = 2.576). All procedures were performed in accordance with the ethical standards of the institutional and national research committee and of the Helsinki declaration and its later amendments or comparable ethical standards. All participants provided written informed consent for participation in this study.

The research project was publicized through regional journals, social media platforms, and flyers distributed in traditional shops (e.g., coffee shops, bakeries, and hair saloons). Personalized invitations to participate in the research project were also extended over the phone using data from the city council of Bragança. Potential participants were told about the objectives of the study, as well as the voluntary nature of participation and the potential physical harms that could occur during assessments. An inclusive strategy was adopted to recruit as many potential participants from the community as possible. All participants who met the inclusion criteria and chose to participate voluntarily were invited to participate in this study. For objective and safety reasons, the following inclusion requirements had to be met: participants must be aged 65 years or older, have the ability to stand and walk with or without assistive equipment, be physically active, and live in the community. Those with chronic neuromuscular, cardiovascular, or metabolic conditions that could present danger or a safety risk during assessment were not considered, and individuals who did not satisfy the above criteria were excluded from participating for safety reasons.

All testing for a given subject was performed on the same day and took approximately 2 h to complete, which included breaks to avoid fatigue. Upon entering the laboratory, participants had their height measured using a wall-mounted tape measure and their weight recorded using a weight scale. All test procedures were explained prior to the assessments. The order in which outcomes were measured depended on station availability.

### 2.3. Measurement Outcomes

A 30 s Chair Stand Test was used for measuring lower body muscle strength [24]. This test measures the number of times an individual can stand up from a chair and sit back down in 30 s. The researcher explained the test to the participant and demonstrated suitable standing and sitting techniques on the chair. The participant was instructed that the goal is to stand and sit as many times as possible in 30 s. The participant was requested to sit on the chair with their back straight and their feet flat on the floor. The researcher explained to the participant that they should stand up from the chair and sit back down as many times as they could in 30 s, without using their arms to push off the chair. The stopwatch started after the verbal cue “Ready, go”. The participant had to stand up completely, and then, sit back down completely before standing up again. The participant had to continue to stand up and sit down as many times as they could in 30 s. At the end of the 30 s, the researcher said “Stop”, and recorded the number of times the participant stood up and sat down completely. The test was repeated up to two times, with the highest score used for analysis.

A Timed Up-and-Go test was used for measuring agility [24]. The time it took an individual to stand up from a chair, walk 2.44 m, turn around, walk back to the chair, and sit down is measured using this test. The participant was asked to sit on the chair with their back straight and their feet flat on the floor. The researcher explained to the participant that they would stand up from the chair, walk a short distance, turn around, and sit back down in the chair. The stopwatch started after the verbal cue “Ready, go”. The participant had to stand up from the chair, walk a distance of 2.44 m, turn around, and walk back to the chair. The participant had to sit back down on the chair. The stopwatch was stopped when the participant’s back was in contact with the back of the chair. This test was measured to the closest 1/10th of a second. The risk of falling was measured using the Berg Balance Scale [25]. This is a 14-item measure (item description example: “sitting to standing”) with each item consisting of a five-point scale ranging from 0 (“lowest level of function”) to 4 (“highest level of function”). The instructions and goal of the test were provided to each participant.

The fear of falling was measured using the Fall Efficacy Scale—International [26]. The FES-I is a 10-item test (item example: “take a bath or shower”) that is rated on a 10-point scale from 1 (“not confident at all”) to 10 (“completely confident”).

### 2.4. Statistical Analyses

Mean, standard deviation, and normal distribution analyses were performed. Cutoffs for normality were determined, with scores of −2/+2 and −7/+7 for skewness and kurtosis, respectively, representing normal distribution. In addition, bivariate Pearson correlations of the variables under analysis were evaluated. The significance level was set at *p* < 0.05. These analyses were conducted using IBM SPSS Statistics version 23.

To examine the associations between variables of interest, analyses were performed in Mplus version 7.4 [27]. We used the robust maximum likelihood estimator to correct any non-normality bias. Due to a priori model saturation, we followed procedures proposed by Hair et al. [28], conducting path analysis with observable variables. Thus, common goodness-of-fit indices to assess model fit were not considered. Direct and indirect effects were analyzed according to standardized coefficients and their respective 95% confidence intervals (CI 95%). Regression paths were considered significant if the CI 95% did not include zero [29]. The *R*^2^ value was calculated to obtain the corresponding effect size, with scores considered trivial at 0.00–0.19, small at 0.20–0.49, average at 0.50–0.79, and large at ≥0.80.

## 3. Results

The descriptive statistics and the correlation matrix are reported in Table 1. The results display no violations of normal distribution, since skewness and kurtosis are contained within the cutoff range. The results revealed that muscle strength was positively correlated with agility. Agility was positively and significantly correlated with fear of falling and negatively correlated with risk of falling. The association between risk and fear of falling was negative and significant.

The proposed model showed that muscle strength was negatively associated with agility (see Figure 1). Consequently, agility was negatively associated with fear of falling. The same trend appeared between fear and risk of falling. The indirect effects were all significant, in which muscle strength displayed a significant positive indirect effect on fear of falling and a negative indirect effect on risk of falling. Agility displayed a significant positive indirect effect on risk of falling. The effect sizes varied between medium (for fear of falling (*R*^2^ = 0.29)) and small (for agility (*R*^2^ = 0.16) and risk of falling (*R*^2^ = 0.03)).

## 4. Discussion

The purpose of this study was to evaluate the effects of muscle strength, agility, and risk of falling on fear of falling in older adults. As hypothesized, lower fear of falling and agility mediated the relationship between muscle strength and risk of falling. Although muscle strength did not directly influence fear of falling, poor agility, and higher fear of falling predicted greater scores for risk of falling in older adults.

The participants in this study showed better results for muscle strength and agility compared to the cutoff values described in existing research [24,26,30,31]. This could have been influenced by the activities performed by the elderly people, since it has been reported that physically active older adults display greater levels of physical fitness [4,32,33]. Thus, it might be suggested that active aging could play a major role in physical capability and function towards daily tasks in the elderly. In fact, the mean score for fear of falling was above the cutoff, suggesting that the participants in this study had greater confidence in their ability to efficaciously and autonomously perform different activities.

Older adults should be able to shift their effort to the limits of their base of support. However, the maximal activation of muscle strength is constrained by various factors, including perceived capability to shift muscle activation towards the muscles of the lower extremities. Based on measures of isokinetic strength, Monteiro et al. [23] suggested that older adults with greater muscle strength have higher dynamic and static balance, which was related to lower perceived risk of falling. While postural instability and imbalance might be the obvious targets for intervention by exercise physiologists in this group of older adults, the current results suggest that directly assessing agility when performing predefined tasks, as well as perceived steadiness and efficacy, to overcome fear of falling is a necessary additional focus. In addition, attention should be paid to the increase in muscle strength of the lower extremities, as this was also a significant determinant in the level of risk of falling reported. Our results provide support to the existing literature [4,16], showing that greater lower body muscle strength was significantly related to agility, and consequently, fear of falling. The validity of the proposed model in this study is supported by a previous study [23] that identified lower limb muscle strength as a predictor of gait parameters.

The present study provides guidance on how to intervene to prevent or reduce falls. First, improving muscle strength in the elderly might decrease fear and, consequently, risk of falling. Forte et al. [22] showed that increased muscle strength improved health and reduce falls in older adults. Using multicomponent training, Monteiro et al. [34] concluded that lower physical fitness was a predictor of risk of falling in older adults. Given the significant contribution of physical fitness to health-related fitness, physical interventions might prove to be especially effective in preventing falls in older adults. By employing a physical training regime along with dual-task activities, a potentially effective fall-prevention program could be developed. There is also reason to believe that greater awareness of exposure and avoidance behaviors related to fall situations (e.g., walking with low room visibility) would be helpful in developing successful fall prevention programs [35].

While it was not measured, there is evidence that cognitive ability could be associated with risk of falling. The relationships between cognitive ability, fear of falling, and balance ability in older adults are complex. Age-related cognitive decline can negatively affect balance ability, leading to increased fear of falling [36]. Additionally, fear of falling can further contribute to decreased balance ability, as individuals may limit their physical activity and become less physically active [37]. Research has shown that older adults with better cognitive abilities have better balance ability and are less likely to experience falls [35]. Cognitive training and physical exercise interventions have also been shown to improve balance ability and reduce fear of falling in older adults [38]. Therefore, maintaining cognitive function may be critical to improving balance ability and reducing fear of falling in older adults.

A contribution of the present study is that detectable lower scores for balance were evident in older adults identified as having greater scores for fear of falling. This supports existing literature suggesting that physical fitness has predictive power toward psychological outcomes [17,19]. However, greater physical fitness was a significant indirect predictor of lower risk of falling. The associations between balance and lower extremity muscle strength further support these assumptions [4]. The restoration of balance confidence may be encouraged by increasing physical activity levels and functions that often accompany fear of falling.

### Limitations, Strengths, and Agenda for Future Research

Several limitations can be found in the present study. While statistically significant, the present study sample consisted of a limited number of participants who were recruited through a convenience sampling method, thus limiting the generalization of the findings to other populations with different cultural backgrounds. In addition, the present cross-sectional study was unable to establish causality. Thus, future studies should test the proposed model using larger samples while considering different experimental study designs. Additionally, the participants in this study were active adults who engaged in physical activity programs. Their levels of physical fitness could have influenced the current results, as sedentary older adults may show lower scores for muscle strength and agility. Future studies should examine the associations among physical fitness and fall-related measures in physically active and sedentary older adults. Moreover, we did not measure falls that participants had experienced in the past, or whether cognitive ability has an impact on falls. Forthcoming studies should assess this variable as it could be the main outcome, and they should also measure cognitive ability as it could also influence risk and fear of falling.

## 5. Conclusions

The main finding of the present study was that muscle strength was significantly correlated with agility, which, in turn, predicted fear of falling. Consequently, lower scores for fear of falling explained lower risk of falling in community-dwelling older adults. Specifically, while muscle strength is a crucial component of physical fitness, only with adequate levels of agility and balance can older adults possess the ability to perform daily tasks. Thus, exercise programs composed of strength and agility training, as well as balance training, appear to be important components in developing a greater ability to walk autonomously, in order to improve functional capacity and lower the probability of falling in the elderly.

## Figures and Tables

**Figure 1 ijerph-20-04945-f001:**
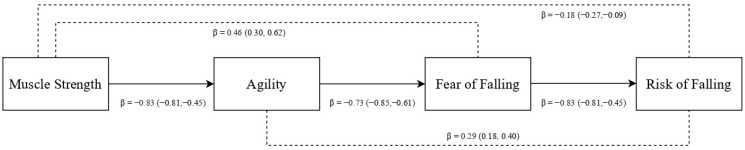
Path model. Notes: Standardized coefficients are reported; within brackets = 95% confidence interval; straight lines = direct effects; doted lines = indirect effects.

**Table 1 ijerph-20-04945-t001:** Descriptive statistics and correlations.

Variables	Unit	Range	M	SD	S	K	1.	2.	3.	4.
1. Chair Stand Test	Repetitions	11–35	20.90	5.82	0.51	−0.11	1			
2. Timed Up-and-Go Test	Seconds	3.29–7.10	4.65	0.81	0.94	0.86	−0.68 **	1		
3. Berg Balance Scale	Points	41–52	47.32	2.97	−0.59	−0.38	0.23	−0.42 **	1	
4. Falls Efficacy Scale	Points	16–29	19.89	3.66	0.87	−0.20	−0.23	0.36 **	−0.40 **	1

Notes: M = mean; SD = standard deviation; S = skewness; K = kurtosis; ** *p* < 0.01.

## Data Availability

Data are available upon request to the authors.

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
