# Peer review of "Effects of Muscle Strength, Agility, and Fear of Falling on Risk of Falling in Older Adults"

_ijerph, 2023, doi:10.3390/ijerph20064945_

Round 1
Reviewer 1 Report
The concept of muscle strength, agility, and fear of falling effects on falling in older adults has been studied in the literature. Besides lack of novelty, there are several incomplete sentences in the manuscript affecting its scientific rigor.
The methods require additional information on instructions for performing functional tests used in the study their validity in the elderly population. The results reflect no information about number of screened participants and their demographics.
Table 1. requires adding units used to perform the chair stand and TUG tests.
Reviewer 2 Report
Inclusion criteria are mentioned in two different sections. Please modify this information to ease the reader which criteria you used when selecting the sample to study. Moreover, where were this population recruited?
Also, please describe in your methods the outcomes measured and which test were used for each of them. This is not indicated on the study. Finally, final sample size is not indicated on methods but in your results instead. If you registered demographic and social data about your participants, please include a table 1 with this information, so the reader can have an idea of the characteristics of your sample.
Discussion is well written, however some changes on the English writing have to be done.
Reviewer 3 Report
Falls in the elderly are an important issue for the independence of ADL and the improvement of QOL of the elderly. This study examines the relationship between muscle strength, agility, and fear of falling in the elderly and the risk of falling. The reviewer believes that the authors need to modify and correct the following points.
1. Regarding the method, â‘ which muscle group is the result of measuring of muscle strength and agility? â‘¡It is necessary to describe in detail the measurement method of â‘ . In particular, motion analysis in agility is important for the discussions. â‘¢ Tests related to balance ability are listed in Table 1, but it is not described in the method.
2 Regarding the results, ①what does Figure 1 show? Which sentence is the results for Table 1?
3 Regarding the discussion, there is a description that "The novel finding in the present study is that detectable lower scores in balance ability were evident in older adults identified with lower scores of being fearful of falling." What research do you think is needed to support the sentence of your opinion? For example, the decline of cognitive ability in older adults may lead to lower scores of the fear of falling, as well as lower balance ability. However, this paper does not measure cognitive ability in the elderly. the author's other study and other literature is needed to support the authors' opinions.
Round 2
Reviewer 2 Report
Paper ready to be published.
Author Response
Thank you for your feedback.
Reviewer 3 Report
The author has not reflected in the answer and the paper regarding the following addition and correction hand number 3. (See sentence number (3) below.)
That is to say, the literatures on the relationship between cognitive ability, fear of falling, and balance ability in the older adults should be cited. Because there are many research papers about these relationships. Finally, the author should clearly state that cognitive ability was not evaluated in this experiment, and discuss the limitations of the study.
(3) With regard to the discussion, "We have a new finding that older adults with lower fear of falling have lower balance ability scores." The statement that this can be proven by
What should be done? For example, an elderly person's cognitive decline could result in a lower fear score, as well as a lower ability to balance. However, this paper does not measure cognitive tests in the elderly. Literature is needed to support the author's discussion.
Author Response
27th February 2023
RE: Effects of muscle strength, agility, and fear of falling on the risk
of falling in older adults
My colleagues and I would like to thank you for the opportunity to resubmit our manuscript to Medicina. We found reviewers’ comments to be very helpful, and we have done our best to incorporate all their suggestions. We thank them for the careful and insightful review of our manuscript. We believe that this has made a significant contribution to the overall quality of the manuscript.
We address all of the concerns of the reviewers in this cover letter. We have also included an updated version of our manuscript with all the changes highlighted using the track-change option in MS WORD. If you require any additional information, please do not hesitate to get in touch with us.
Sincerely,
Pedro Morouço (PhD)
Reviewer 1
The author has not reflected in the answer and the paper regarding the following addition and correction hand number 3. (See sentence number (3) below.)
Response: We apologize for not having clearly revised your comment in the manuscript. We believe that in this revised version we have done our best to incorporate your suggestions. Point-by-point responses as provided and revisions are tracked in the manuscript using the track change option in MS Word.
That is to say, the literatures on the relationship between cognitive ability, fear of falling, and balance ability in the older adults should be cited. Because there are many research papers about these relationships.
Response: We consider that our discussion on cognitive ability goes beyond the objective of the present study. Thus, we revised our discussion and focused on the physical fitness of older adults.
Finally, the author should clearly state that cognitive ability was not evaluated in this experiment and discuss the limitations of the study.
Response: Limitation inserted.
(3) With regard to the discussion, "We have a new finding that older adults with lower fear of falling have lower balance ability scores." The statement that this can be proven by. What should be done? For example, an elderly person's cognitive decline could result in a lower fear score, as well as a lower ability to balance. However, this paper does not measure cognitive tests in the elderly. Literature is needed to support the author's discussion.
Response: We consider that our discussion on cognitive ability goes beyond the objective of the present study. Thus, we revised our discussion and focused on the physical fitness of older adults.